# Altered Nucleotide Insertion Mechanisms of Disease-Associated TERT Variants

**DOI:** 10.3390/genes14020281

**Published:** 2023-01-21

**Authors:** Griffin A. Welfer, Veniamin A. Borin, Luis M. Cortez, Patricia L. Opresko, Pratul K. Agarwal, Bret D. Freudenthal

**Affiliations:** 1Department of Biochemistry and Molecular Biology, University of Kansas Medical Center, Kansas City, KS 66103, USA; 2University of Kansas Cancer Center, Kansas City, KS 66103, USA; 3Department of Physiological Sciences and High-Performance Computing Center, Oklahoma State University, Stillwater, OK 74077, USA; 4Department of Cancer Biology, University of Kansas Medical Center, Kansas City, KS 66103, USA; 5Department of Environmental and Occupational Health, University of Pittsburgh Graduate School of Public Health, and UPMC Hillman Cancer Center, Pittsburgh, PA 15232, USA

**Keywords:** telomeres, telomerase, telomere biology disorders, cancer, enzyme mechanisms

## Abstract

Telomere biology disorders (TBDs) are a spectrum of diseases that arise from mutations in genes responsible for maintaining telomere integrity. Human telomerase reverse transcriptase (hTERT) adds nucleotides to chromosome ends and is frequently mutated in individuals with TBDs. Previous studies have provided insight into how relative changes in hTERT activity can lead to pathological outcomes. However, the underlying mechanisms describing how disease-associated variants alter the physicochemical steps of nucleotide insertion remain poorly understood. To address this, we applied single-turnover kinetics and computer simulations to the *Tribolium castaneum* TERT (tcTERT) model system and characterized the nucleotide insertion mechanisms of six disease-associated variants. Each variant had distinct consequences on tcTERT’s nucleotide insertion mechanism, including changes in nucleotide binding affinity, rates of catalysis, or ribonucleotide selectivity. Our computer simulations provide insight into how each variant disrupts active site organization, such as suboptimal positioning of active site residues, destabilization of the DNA 3′ terminus, or changes in nucleotide sugar pucker. Collectively, this work provides a holistic characterization of the nucleotide insertion mechanisms for multiple disease-associated TERT variants and identifies additional functions of key active site residues during nucleotide insertion.

## 1. Introduction

Human chromosomes are capped by nucleoprotein structures called telomeres that prevent chromosome ends from undergoing degradation and inappropriate recombination [1,2]. However, the DNA polymerases that replicate the genome cannot completely copy the lagging strand, which causes telomeres to shorten with each round of DNA replication [3]. Critically shortened telomeres lose their protective function and trigger a DNA damage response that results in either replicative senescence, or apoptosis [4,5]. Telomere shortening is counteracted by the ribonucleoprotein telomerase, which synthesizes de novo telomeric repeats at chromosome ends [6]. Telomerase is transiently expressed in highly proliferative tissues such as stem and germ cells, but most somatic cells silence it following differentiation [7]. Furthermore, over 85% of cancers reactivate telomerase to achieve replicative immortality [7,8,9,10]. Conversely, insufficient telomerase activity causes premature telomere shortening and is associated with a spectrum of diseases called telomere biology disorders (TBDs). Common TBDs include aplastic anemia (AA), dyskeratosis congenita (DC), and idiopathic pulmonary fibrosis (IPF) [11,12,13,14,15]. Therefore, a precise balance of telomerase activity is required for maintaining long-term genomic integrity without pathological consequences. This association between telomerase and numerous human diseases underscores the importance of understanding the mechanism of this enzyme at the molecular level.

The catalytic core of human telomerase consists of the catalytic protein subunit, telomerase reverse transcriptase (hTERT), and the telomerase RNA component (hTR) (Figure 1A) [16,17]. hTERT synthesizes telomeric repeats by reverse transcribing an internal RNA template within hTR (Figure 1B) [18,19,20]. The catalytic cycle of telomerase begins with the template region of hTR annealing to the 3′ end of a complementary single-stranded telomeric DNA sequence (Figure 1B, step 1) [21]. hTERT then uses the first nucleobase in its template to bind a complementary deoxyribonucleoside triphosphate (dNTP). If the incoming dNTP forms proper Watson–Crick base pairs with the templating base, hTERT will catalyze the formation of a phosphodiester bond between the terminal 3′ hydroxyl group of the DNA strand and the α-phosphate of the incoming dNTP [22]. Following bond formation, hTERT shifts its registry to the next template position to insert another dNTP. Telomerase can undergo six rounds of nucleotide insertion before reaching the end of its template sequence, and its ability to consecutively add nucleotides is determined by its nucleotide insertion processivity (NAP) (Figure 1B, step 2) [23]. After synthesizing a complete repeat, the entire telomerase complex can translocate along the newly extended DNA strand to realign the DNA 3′ terminus with the beginning of its RNA template for another round of synthesis [24]. This process of translocation and extension is known as repeat addition processivity (RAP) (Figure 1B, step 3) [25]. After a variable number of translocation cycles, telomerase will end its catalytic cycle by completely dissociating from the telomere (Figure 1B, step 4). 

Telomerase activity is primarily measured using direct primer extension assays [26,27]. These assays are typically performed by incubating telomerase with a radiolabeled single-stranded telomeric DNA oligonucleotide and dNTPs to monitor processive telomere extension [28]. The reaction products are resolved on a sequencing gel and the total number of synthesized repeats is quantified [17,26,29,30]. Recent global kinetic analyses of telomerase suggest that template translocation associated with RAP is the rate-limiting step of its catalytic cycle [29,31]. Therefore, steady-state assays such as the direct primer extension assay are useful for evaluating general activity and RAP differences between variants but do not provide information into TERT’s mechanism of nucleotide insertion [32]. Various model systems have been used to study telomerase’s catalytic mechanism [33,34,35]. In this study, we used the tcTERT model because it shares a high degree of sequence identity with hTERT’s active site (Appendix A) [22]. Furthermore, the atomic resolution structures of tcTERT bound to an incoming dNTP have been solved, which allows for rigorous computational characterization of its nucleotide insertion mechanism [36]. tcTERT consists of three domains: the telomerase RNA binding domain (TRBD), the reverse transcriptase (RT) domain, and the C-terminal extension (CTE) domain, all of which overlay well with hTERT (Appendix A) [19,33,37]. tcTERT lacks the telomerase essential N-terminal (TEN) domain and the IFD-TRAP motif found in other TERTs [38]. TEN and IFD-TRAP promote telomerase recruitment and RAP [39,40]. Therefore, we used tcTERT for pre-steady-state kinetic analysis of the nucleotide insertion mechanisms of disease-associated TERT variants. By investigating individual nucleotide insertion events, we can determine the altered nucleotide insertion mechanisms of disease-associated variants independently of telomerase recruitment and RAP [41,42]. 

Here, we use single-turnover kinetics (STK) and molecular dynamics (MD) simulations to investigate the mechanisms of nucleotide insertion for six disease-associated TERT variants. The variants investigated were (tcTERT^hTERT^) R194Q^R631Q^, A255V^A716V^, Y256N^Y717N^, R340H^R865H^, V342M^V867M^, and K372N^K902N^. For clarity, tcTERT residue numbering will be used throughout the text, and homologous hTERT residues can be found in (Table 1). These residues cluster into three groups: (1) R194 and K372 interact with the triphosphate of the incoming nucleotide, (2) R340 and V342 interact with the DNA 3′-terminus, and (3) A255 and Y256 interact with the sugar of the incoming nucleotide (Figure 1C and Appendix A). Collectively, we measured the catalytic efficiencies of each variant and identified interactions responsible for disrupting TERT activity.

## 2. Materials and Methods

### 2.1. Nucleic Acid Sequences

Nucleic acids were purchased from Integrated DNA Technologies (IDT). Single-turnover kinetic experiments were performed with 5′-6-carboxyfluorescein (6-FAM) labeled DNA primers (5′-CCAGCCAGGTCAG-3′) and unlabeled RNA primers (5′-rUrGrArCrCrUrGrArCrCrUrGrGrCrUrGrG-3′). Oligonucleotides were resuspended in molecular biology grade water and concentrations were determined by measuring absorbance at 260 nm with a NanoDrop One UV–Vis Spectrophotometer (Thermo Scientific, Waltham, MA, USA). DNA:RNA duplexes were annealed at a 1:1.2 molar ratio by heating the samples to 90 °C for 2 min, followed by cooling them to 4 °C at a rate of 0.1 °C per second using a thermocycler.

### 2.2. Expression and Purification of tcTERT

The full-length *T. castaneum* TERT gene was purchased from GenScript and cloned into a modified pET28b vector containing a Tobacco Etch Virus (TEV) protease cleavable N-terminal hexahistidine tag. Each tcTERT variant was generated using the Q5^®^ Site-Directed Mutagenesis Kit (New England Biolabs, Ipswich, MA) and confirmed by sanger sequencing. tcTERT was expressed in BL-21 (DE3) pLysS cells and grown in an Epiphyte3 LEX bioreactor at 37 °C until reaching an OD600 of 0.6. The temperature was then dropped to 30 °C and Isopropyl β-D-1-thiogalactopyranoside (IPTG) was added to a final concentration of 0.4 mM to induce tcTERT expression overnight. Cells were harvested the following morning (approximately 16 h post-induction) by centrifugation at 4200× *g* and lysed via sonication in a buffer containing 50 mM TRIS (pH 7.5), 10% glycerol, 0.75 M KCl, 5 mM 2-mercaptoethanol, and 1 mM each of the following protease inhibitors: 4-(2-aminoethyl)benzenesulfonyl fluoride hydrochloride (AEBSF), leupeptin, pepstatin A, and benzamidine.

For purification of tcTERT, 50 mM imidazole (pH 7.5) was added to the lysate prior to loading over a Ni-NTA column (GE Healthcare) that had been equilibrated with a buffer consisting of 50 mM TRIS (pH 7.5), 10% glycerol, 0.75 M KCl, and 5 mM 2-mercaptoethanol. tcTERT was eluted using a 0–100% gradient of a buffer consisting of 50 mM TRIS (pH 7.5), 10% glycerol, 0.75 M KCl, 5 mM 2-mercaptoethanol, and 500 mM imidazole. Fractions containing tcTERT were diluted by ~50% using a buffer consisting of 50 mM TRIS (pH 7.5), 10% glycerol, 0.5 M KCl, and 5 mM 2-mercaptoethanol before being loaded over a POROS HS column (Thermo Fisher, Waltham, MA, USA) equilibrated with a buffer consisting of 50 mM TRIS (pH 7.5), 10% glycerol, 0.5 M KCl, and 5 mM 2-mercaptoethanol. tcTERT was eluted with a 0–100% gradient of a buffer consisting of 50 mM TRIS (pH 7.5), 10% glycerol, 5 mM 2-mercaptoethanol, and 1.5 M KCl. Following elution, the hexahistidine tag was cleaved overnight at 4 °C using TEV protease (at a concentration of 1 mg TEV protease per 10 mg tcTERT) and tcTERT with the tag cleaved off was purified away from uncleaved protein with another run over the Ni-NTA column as described above. Monodisperse tcTERT was purified using size exclusion chromatography on a Sephacryl S-200 16/60 column (GE Healthcare) using a buffer consisting of 50 mM TRIS (pH 7.5), 10% glycerol, 0.8 M KCl, and 1 mM Tris(2-carboxyethyl)phosphine (TCEP). Finally, a buffer exchange was performed on the pooled fractions containing tcTERT so that the final buffer consisted of 5 mM TRIS (pH = 7.5), 10% glycerol, 0.5 M KCl, and 1 mM TCEP. tcTERT was then concentrated to between 16–22 mg mL^−1^ using 10 K centrifugal filters (Millipore Sigma, Burlington, MA, USA) and either aliquoted and snap-frozen in liquid nitrogen or used in downstream experiments.

### 2.3. tcTERT Single-Turnover Kinetics

Single-turnover kinetics were performed using a Kin Tek RQF-3 rapid quench flow instrument as described previously [50]. Prior to performing the reactions, 2 μM tcTERT was incubated with 200 nM pre-annealed DNA:RNA hybrid oligonucleotide in a buffer containing 50 mM TRIS (pH 7.5), 200 mM KCl, 10% glycerol, 2 mM dithiothreitol (DTT), and 0.1 mg mL^−1^ Bovine Serum Albumin for 30 min on ice. Following equilibration, the tcTERT:DNA:RNA complex was mixed with equal volumes of a solution containing 10 mM MgCl_2_, 200 mM KCl, 10% glycerol, and various concentrations of nucleotide triphosphate to initiate each reaction. Each reaction was performed at 37 °C and quenched at various timepoints with a solution containing 200 mM EDTA (pH 8.0). The products of each reaction were transferred into a solution of DNA loading buffer consisting of 100 mM EDTA, 80% deionized formamide, 0.25 mg mL^−1^ bromophenol blue, and 0.25 mg mL^−1^ xylene cyanol. Datasets with a minimum time point of 12 s or greater were acquired using a LabDoctor heating block. The reaction conditions were the same as outlined above; however, each reaction was directly quenched with a solution of DNA loading buffer. The products of each experiment were incubated at 95 °C for 5 min, loaded onto a 21% denaturing urea-polyacrylamide gel, and ran at 700 V/60 A/30 W at 30 °C until single nucleotide insertions could be resolved.

Following electrophoresis, the gels were scanned with a GE Typhoon, FLA 9500 imager. The images were analyzed in ImageJ, and the ratio of product to substrate was quantified. The means and standard deviations from at least three replicates were plotted using GraphPad Prism (Version 9.3.1), and reaction progress plots were fit to a single exponential equation (Equation (1)) using nonlinear regression:(1)$${\left[P\right]}_{t}=A\left(1-e^{-kt}\right)$$
where ${\left[P\right]}_{t}\;$is the concentration of product at time *t*, A is the amplitude of the burst phase and is dependent on the concentration of tcTERT:DNA:RNA complexes at the start of the reaction, and *k* is the pseudo-first-order rate (*k_obs_*) constant for nucleotide insertion at the concentration of nucleotide used in the reaction. Quantification of reaction products and full kinetic parameters for each curve used to derive *k_obs_* and A values are provided in (Appendix A). *k_obs_* values were derived using increasing nucleotide concentrations and fit to a hyperbolic function (Equation (2)) using nonlinear regression with GraphPad Prism (Version 9.3.1).
(2)$$k\left(\mathrm{dNTP}\right)=\frac{k_{pol}\left[d\mathrm{NTP}\right]}{K_{d}+\left[d\mathrm{NTP}\right]}$$
where *k*(dNTP) is the *k_obs_* value for a given dNTP concentration, *k_pol_* is the maximum rate constant for a polymerase inserting the given dNTP, [dNTP] is the concentration of the nucleotide of interest, and *K_d_* is the apparent equilibrium dissociation constant for the specific nucleotide. *k_pol_* and *K_d_* were calculated using (Equation (2)), and the standard error of the mean for each parameter is reported.

### 2.4. Computer Simulations

Molecular dynamics (MD) simulations were performed for tcTERT and DNA complexes in explicit water solvent, using a protocol similar to our previous study [51]. Model preparation and simulations were performed using the AMBER v16 suite of programs for biomolecular simulations [52]. AMBER’s *ff14SB* [53] force-fields were used for all simulations. MD simulations were performed using NVIDIA graphical processing units (GPUs) and AMBER’s *pmemd.cuda* simulation engine using our lab protocols published previously [54,55]. We have verified the suitability of ff14SB for simulations at microsecond timescales for a number of proteins and protein-DNA complexes [51,56,57]. Standard parameters from ff14SB force-field were used for the protein residues and nucleotides. Non-standard nucleotides (dGTP and rGTP) were parameterized using procedures described in the AMBER manual. SPC model was used for water. The parameters for ions were used from AMBER, as available in the *frcmod.ionsjc_spce* and *frcmod.ionslrcm_hfe_spce* files. 

A total of 12 separate simulations were performed for TERT ternary complexes based on the X-ray crystal structure of wild type tcTERT bound to an incoming nucleotide (PDB: 7KQN). In addition to wild type tcTERT, simulations with the following mutations R194Q, A255V, Y256A, Y256N, R340H, V342M, and K372N were performed with dGTP. Additionally, simulations of wild type, A255V, Y256A, and Y256N variants were performed with rGTP. Three separate simulations for binary complexes (tcTERT bound to a DNA:RNA oligonucleotide) were performed for wild type, R340H, and V342M. Starting from the wild type structure, the variants were computationally created by deleting the side-chain atoms of the mutated residue and adding the new chain in the *tleap* module of AMBER. The missing hydrogen atoms were also added by AMBER’s *tleap* program. After processing the coordinates of the protein and substrate, all systems were neutralized by addition of counter-ions, and the resulting systems were solvated in a rectangular box of SPC/E water, with a 10 Å minimum distance between the protein and the edge of the periodic box. The prepared systems were equilibrated using a protocol described previously [58,59]. The equilibrated systems were then used to run 1.0 μs of production MD under constant energy conditions (NVE ensemble). The use of NVE ensemble is preferred as it offers better computational stability and performance [60,61]. The production simulations were performed at a temperature of 300 K. As NVE ensemble was used for production runs, these values correspond to the initial temperature at the start of simulations. Temperature adjusting thermostat was not used in simulations; over the course of 1.0 μs simulations, the temperature fluctuated around 300 K with RMS fluctuations between 2–4 K, which is typical for well-equilibrated systems. A total of 1000 conformational snapshots (stored every 1 nanosecond) were collected for each system and used for analysis. The distances and percentage occupancy were calculated using the 1000 conformations stored during the MD simulations. A cutoff of 3.4 Å was used to calculate the percent of snapshots within hydrogen bonding distance (occupancy) during each MD simulation.

## 3. Results

### 3.1. R194Q and K372N Disrupt the Electrostatic Environment of the dNTP’s Triphosphate

R194 extends from above TERT’s active site into the hydrogen bonding distance of the incoming nucleotide’s triphosphate (Figure 2A) [22]. This positioning suggests it could play a role in facilitating productive nucleotide binding [22,62,63]. The R194Q TERT variant is associated with IPF, a lethal disease that affects lung parenchyma [43,44]. We determined the altered mechanism of R194Q nucleotide insertion using STK and MD simulations. Using STK, we determined the apparent equilibrium dissociation constant for the dNTP binding step (*K_d_*) and the maximal rate constant for nucleotide insertion (*k_pol_*) Appendix A) [32]. The ratio (*k_pol_ K_d_*^−1^) defines TERT’s catalytic efficiency and is a directly comparable quantity for assessing the efficiency of nucleotide insertion between variants [32]. R194Q inserted dGTP across from a templating rC with a *K_d_* of 161 ± 8 μM, which is 9-fold higher than the previously reported value for WT tcTERT (*K_d_* = 18 μM) [22]. Additionally, R194Q had a *k_pol_* of 0.197 ± 0.004 s^−1^, which is 5-fold slower than the previously reported value for WT tcTERT (*k_pol_* = 1.05 s^−1^) [22] (Figure 2B). These changes in *k_pol_* and *K_d_* decrease the catalytic efficiency of R194Q 47-fold compared to the previously reported efficiency for WT tcTERT (Table 2) [22].

Along with STK, we performed 1 μs MD simulations with WT and R194Q ternary complexes (TERT:DNA/RNA:dGTP). The WT TERT ternary simulation revealed R194 transiently hydrogen bonds with different oxygens of the triphosphate in 80% of conformations sampled (Appendix A). In addition to its interactions with the triphosphate, R194 hydrogen bonds with the sidechain of Q308 (Q833 in hTERT) above the nucleobase of dGTP in 24% of conformational snapshots (Appendix A). In the R194Q ternary complex, Q194 only interacted with the triphosphate of dGTP in 1% of conformational snapshots (Appendix A). Yet, Q194 hydrogen bonds with the side chain of Q308 during 85% of conformations (Appendix A). Interestingly, when we simulated the WT TERT binary complex (TERT:DNA/RNA), R194 hydrogen bonded with Q308 during 85% of conformations (Appendix A). This suggests that R194 interacts with Q308 to organize TERT’s active site prior to nucleotide binding; and, upon formation of the ternary complex, R194 stabilizes the electrostatics of the triphosphate. Previous kinetic analysis of R194A TERT determined that it has a *k_pol_* of 0.037 s^−1^, which is 5-fold slower than R194Q [22]. However, R194A had a *K_d_* of 93 μM, which is less than 2-fold different than R194Q. These parameters suggest the tandem contributions of R194 with the incoming nucleotide’s triphosphate and Q308 are required for maximal activity. R194’s interaction with the triphosphate is the dominant factor determining nucleotide binding affinity since both R194Q and R194A have similar *K_d_* values. Furthermore, R194’s interactions with the triphosphate and Q308 appear to contribute equally to the rate of nucleotide insertion since R194Q had a 5-fold decrease in *k_pol_* with respect to WT upon losing the triphosphate interaction, and R194A had a further 5-fold decrease in its *k_pol_* with respect to R194Q (25-fold less than WT) upon losing its interaction with Q308 [22]. This is consistent with a mechanism where R194 stabilizes the electrostatic charge of the dNTP’s triphosphate, while simultaneously stabilizing the dNTP’s nucleobase.

K372 extends from above the DNA 3′ terminus to hydrogen bond with nonbridging oxygens on the α and γ phosphates of the incoming dNTP (Figure 2A). K372N dramatically decreases human telomerases catalytic activity and is associated with DC, IPF, and cancer [13]. DNA polymerases catalyze the formation of a phosphodiester bond via hydrolysis of a dNTP at its α-phosphate to produce a dNMP and pyrophosphate [64]. The overall reaction requires two independent proton transfer reactions [65]. First, a catalytic metal ion facilitates deprotonation of the terminal nucleotides 3′-hydroxyl group to generate the nucleophile needed to attack the α-phosphate of the incoming dNTP. Following nucleophilic attack of the dNTP’s α-phosphate by the activated 3′-hydroxyl, the second proton transfer reaction involves protonation of Oαβ on the pyrophosphate leaving group in the polymerase active site. Most DNA polymerases use an active site amino residue for this second reaction, and K372 has been identified as the putative general acid residue for TERT [65,66]. K372N retains some polar characteristics but has no acidic protons for the protonation of Oαβ. Therefore, we sought to characterize K372N nucleotide insertion to provide evidence for K372′s role during TERT nucleotide insertion. 

We determined that K372N inserted dGTP across from a templating rC with a *K_d_* of 178 ± 8 μM, a 10-fold increase compared to WT (*K_d_* = 18 μM) [22]. However, K372N’s *k_pol_* for dGTP insertion was 1.1 × 10^−2^ ± 1.6 × 10^−4^ s^−1^, which is 95-fold slower than WT (*k_pol_* = 1.05 s^−1^) (Figure 2C) [22]. This causes K372N to have a 939-fold reduction in its catalytic efficiency for dGTP insertion (Table 2). To obtain additional mechanistic insight into the role of K372, we performed MD simulations with the K372N ternary complex. Our simulations revealed that in WT TERT, the polar protons of K372 are an average of 3.2 ± 0.5 Å away from Oαβ (Figure 2D,E).

However, in the K372N simulation, the polar protons of N372 are an average of 6.6 ± 1.0 Å away from Oαβ (Figure 2D,E). These distance distributions are likely an upper estimate of the true distances for each residue since Oαβ will become ionized as the reaction progresses, and our simulations only investigated the ternary complexes prior to any chemical steps. The distance between Oαβ and its proton donor directly affects the proton transfer reaction, and increased distance between these two atoms will decrease the rate of the reaction. The increased distance we observed between the polar protons of K372N and Oαβ, along with its 95-fold decrease in *k_pol_*, is consistent with the role of K372 as a general acid during catalysis via proton transfer to Oαβ.

Interestingly, both K372N and R194Q had substantial increases in the number of stable hydrogen bonds with the incoming nucleotide compared to WT. Notably, R194Q and K372N showed compensatory changes in their hydrogen bonding profiles. The R194Q complex formed a hydrogen bond between K372 and a nonbridging oxygen on the γ-phosphate for the entire 1 μs trajectory, and the K372N complex formed a hydrogen bond between R194 and the triphosphate during 89% of conformations. In contrast, WT TERT did not form any stable hydrogen bonds between R194/K372 and the triphosphate. This compensatory behavior is consistent with our kinetics data since R194Q and K372N had identical *K_d_* values and suggests that positively charged active site residues may be partially redundant with respect to their nucleotide binding functions. These results suggest that efficient nucleotide insertion by WT TERT is dependent on continuous dynamic sampling between positively charged active site residues and the incoming nucleotide.

### 3.2. R340H and V342M Disrupt the DNA 3′-Terminus

R340 positions the essential catalytic aspartic acid residues D343 and D344 (D868/D869 in hTERT) in a loop adjacent to the DNA 3′-hydroxyl (Figure 3A), and R340H is associated with IPF [43,49,67,68]. We sought to characterize the mechanism of nucleotide insertion using STK. Because R340 positions residues involved in catalysis, we sought to characterize the mechanism of nucleotide insertion using STK. R340H inserted dGTP across from a templating rC with a *K_d_* of 1337 ± 129 μM, which is 74-fold higher than WT (*K_d_* = 18.1 μM) [22]. Additionally, the *k_pol_* of dGTP insertion for R340H was 0.25 ± 0.01 s^−1^, a 4-fold decrease compared to WT (*k_pol_* = 1.05 s^−1^) (Figure 3B) [22]. This led to a 300-fold decrease in R340H’s catalytic efficiency compared to WT TERT (Table 2). Due to substrate inhibition caused by concentrations of dGTP exceeding 1800 μM R340H reached its maximal activity at subsaturating concentrations of dGTP. 

We next used MD simulations to gain insight into the consequences of R340H on nucleotide insertion. Our simulations revealed the average distance between the 3′-hydroxyl of the terminal nucleotide and the α-phosphate of dGTP for R340H was 5.8 ± 0.3 Å, whereas the average distance for WT TERT was 3.6 ± 0.9 Å (Figure 3C). The deoxyribose sugar of DNA can occupy distinct pseudorotational angles (sugar puckers), where different members of the ring occupy exo or endo conformations [69,70]. Nucleotide sugar pucker has an effect on the catalytic activities of DNA polymerases, with DNA polymerases favoring the C3′-endo conformation, which corresponds to a pseudorotational angle between (0–36°) [71,72]. In our WT simulation, dG3′ occupied two populations centered around 22 ± 15° and 128 ± 49°. These two populations correspond to C3′-endo and C1′-exo sugar puckers, respectively. The sugar pucker for dGTP occupied a single population centered around 32 ± 18°, which corresponds to a C3′-endo conformation (Figure 3D). In our R340H simulation, dG3′ occupied a single population centered around 49 ± 23°, which corresponds to a C4′-exo conformation. The sugar pucker conformations for dGTP in the R340H ternary complex occupied a much broader spectrum than WT with two maxima centered at 16° and 174° (Figure 3D). The broad spectrum of sampled sugar puckers in R340H, along with the increased distance between the 3′-hydroxyl of the terminal nucleotide and the α-phosphate of dGTP, suggests that R340H destabilizes the DNA 3′ terminus and incoming nucleotide compared to WT.

V342 is positioned behind the DNA 3′ terminus with its isopropyl chain making Van der Waals contacts with the deoxyribose sugar of the DNA 3′ terminal nucleotide, and V342M is associated with IPF (Figure 3A) [36,43]. We characterized the nucleotide insertion mechanism of V342M using STK. V342M had a *k_pol_* of 1.9 ± 0.1 s^−1^ for dGTP insertion, which is 2-fold faster than WT (*k_pol_* = 1.05 s^−1^) [22]. However, V342M had a *K_d_* of 126 ± 17 μM, which is 7-fold higher than WT (*K_d_* = 18.1 μM) (Figure 3E) [22]. This means the catalytic efficiency of dGTP insertion for V342M is 4-fold lower than that of WT (Table 2). 

In order to uncover additional mechanistic details that may explain the modest differences in nucleotide insertion efficiency between V342M and WT we used MD simulations to probe its active site dynamics. Our simulations revealed that V342M had more dynamic Watson–Crick hydrogen bonding between the incoming nucleotide and the templating rC. V342M induces the opening of the incoming nucleotides Watson–Crick face, which may increase the *K_d_* for nucleotide binding (Figure 3F). Since V342 sits behind the DNA 3′-terminus and incoming nucleotide, the larger sidechain introduced by V342M likely reduces the volume of the nucleotide binding pocket. The smaller binding pocket would not be able to fully accommodate the incoming nucleotide and thus promotes the opening of the opposite Watson–Crick face (H41-O6) (Figure 3G). Additionally, we calculated the sugar pucker profile of dG3′ and dGTP in the V342M ternary complex. V342M caused a sharpening of the sampled sugar puckers toward the C3′-endo conformation with dG3′ and dGTP populations centered around 16 ± 11° and 23 ± 15° respectively (Figure 3D). Polymerases prefer to insert nucleotides with a C3′-endo conformation, and V342M’s preference for C3′-endo sugar pucker is consistent with its increased *k_pol_* compared to WT TERT. This observed sharpening of the sugar puckers sampled by V342M suggests that its larger sidechain sterically hinders the sugar of both dG3′ and dGTP from sampling conformations.

### 3.3. A255V and Y256N Decrease TERT Sugar Fidelity through Distinct Mechanisms

Y256 is TERT’s steric gate residue because it is positioned directly behind the C2′ position of the incoming dNTP’s sugar moiety, where it sterically clashes with the 2′-hydroxyl of incoming rNTPs to prevent their incorporation into the DNA strand (Figure 4A) [22,73]. Y256N is associated with head and squamous cell carcinoma [48]. Therefore, we sought to determine what effect Y256N had on dGTP insertion using STK. Y256N had a robust rate for dGTP insertion with a *k_pol_* of 6.7 ± 0.4 s^−1^, which is 6-fold faster than WT (*k_pol_* = 1.05 s^−1^) [22]. Y256N also had a *K_d_* of 205 ± 32 μM for dGTP, which is 10-fold greater than WT (Figure 4B). This results in less than a 2-fold difference between the catalytic efficiencies of Y256N and WT (*K_d_* = 18.1 μM) (Table 2) [22]. The role of Y256 as TERT’s steric gate prompted us to investigate Y256N sugar fidelity. Sugar fidelity corresponds to a polymerases ability to discriminate between ribonucleotides and deoxyribonucleotides and is calculated as the ratio of the catalytic efficiency for dNTP insertion to its catalytic efficiency for rNTP insertion [74]. Therefore, to determine the sugar fidelity of Y256N we performed STK using rGTP across from a templating rC. Y256N inserted rGTP at a rate of 0.60 ± 0.03 s^−1^, which is 10-fold slower than its dGTP insertion rate. Y256N had a *K_d_* of 147 ± 18 μM for rGTP, which is nearly the same as its *K_d_* for dGTP insertion (Figure 4C). Consequently, Y256N exhibits an 8-fold preference for dNTPs over rNTPs. However, comparing Y256N sugar fidelity to WT TERT sugar fidelity (*k_pol_* = 3.7 × 10^−3^ s^−1^, *K_d_* = 887 μM) [22] reveals that Y256N has a 157-fold faster *k_pol_* for rGTP and a 7-fold lower *K_d_* for rGTP than WT. This results in a 971-fold increase in the catalytic efficiency of rGTP insertion for Y256N compared to WT (Figure 4D, Table 2).

Y256N had similar binding affinities for both rGTP and dGTP but a significantly reduced catalytic rate for rGTP insertion. To investigate the mechanism underlying Y256N’s decreased insertion rate of rGTP we used MD simulations with Y256N bound to either dGTP or rGTP to compare the effects of each ligand. These simulations revealed that the coordination of the triphosphate is altered for Y256N and WT rGTP complexes (Figure 4E,F). K372 does not interact with the α-phosphate of rGTP and instead coordinates the γ-phosphate of rGTP in an alternate orientation (State 2 Figure 4E). In contrast, when we simulated the Y256A ternary complex, where the steric gate is completely removed and both dNTP and rNTP catalytic efficiencies are the same, we did not see any changes in the triphosphate coordination state (Figure 4F) [22]. This alternative coordination of the triphosphate explains the decreased catalytic rate for rGTP by Y256N.

The flanking mainchain amides of A255 hydrogen bond with the 3′-hydroxyl and a nonbridging oxygen on the β-phosphate of the incoming dNTP (Figure 4A and Figure 5D) [22,36]. The A255V TERT variant is associated with IPF, BMF, and cancer [45,46,47,68]. A255V introduces a bulkier isopropyl group, which could sterically clash with the incoming nucleotide and negatively impact insertion efficiency. Using STK we found A255V inserted dGTP with a *k_pol_* of 0.31 ± 0.01 s^−1^, which is 3-fold slower than WT, and a *K_d_* of 190 ± 23 μM, which is a 10-fold increase over WT (Figure 5A). This results in a 36-fold decrease in its catalytic efficiency for dGTP insertion compared to WT (Table 2). Since A255 is adjacent to the steric gate, we sought to determine the sugar fidelity of A255V. A255V inserted rGTP with a *k_pol_* of 6.6 × 10^−3^ ± 3.9 × 10^−4^ s^−1^, which is 47-fold slower than its *k_pol_* for dGTP. Additionally, A255V had a *K_d_* of 1088 ± 149 μM for rGTP insertion, which is a 6-fold increase compared to its *K_d_* for dGTP (Figure 5B). Comparing A255V and WT rGTP kinetics reveals that both enzymes insert rGTP with nearly the same efficiency. However, A255V’s decreased efficiency for inserting dGTP results in a 270-fold preference for dGTP over rGTP, compared to an 18,000-fold preference in WT (Figure 5C) [22]. Thus, A255V has overall reduced sugar fidelity compared to WT. To investigate the molecular basis of A255V’s decreased sugar fidelity we used MD simulations to investigate the dynamics of dGTP and rGTP within the A255V active site. These simulations showed A255V disrupts the backbone amide hydrogen bonds between the 3′-OH of the incoming nucleotide and the triphosphate (Figure 5E). The backbone amides of WT TERT hydrogen bond with the 3′-OH of dGTP in 34% of conformational snapshots and the nonbridging oxygens of the β-phosphate in 81% of conformations. The same backbone amides did not form any hydrogen bonds with rGTP throughout the simulation. A255V did not form any hydrogen bonds with the 3′-OH of either dGTP or rGTP. However, A255V did hydrogen bond with the nonbridging oxygens of the β-phosphate in 30% of conformations in the dGTP simulation and in 15% of conformations during the rGTP simulation. The decreased hydrogen bonding with the incoming nucleotide observed in the A255V simulation is likely the mechanism of its decreased catalytic efficiency for dGTP insertion. The nearly identical insertion efficiencies for rGTP between WT and A255V suggest that the intact steric gate is the dominant determinant of ribonucleotide insertion efficiency.

**Table 2 genes-14-00281-t002:** Summary of Kinetic Parameters.

tcTERT Variant	Incoming Nucleotide	*k_pol_* (s^−1^)	*K_d_* (μM)	Catalytic Efficiency (μM^−1^ s^−1^)
R194Q	dGTP	0.197 ± 0.004	161 ± 8	1.2 × 10^−3^ ± 6.7 × 10^−5^
A255V	dGTP	0.31 ± 0.01	190 ± 23	1.6 × 10^−3^ ± 2.1 × 10^−4^
A255V	rGTP	6.6 × 10^−3^ ± 3.9 × 10^−4^	1088 ± 149	6.1 × 10^−6^ ± 3.7 × 10^−6^
Y256N	dGTP	6.7 ± 0.4	205 ± 32	3.3 × 10^−2^ ± 5.5 × 10^−3^
Y256N	rGTP	0.60 ± 0.03	147 ± 18	4.1 × 10^−3^ ± 5.2 × 10^−4^
R340H	dGTP	0.25 ± 0.01	1337 ± 129	1.9 × 10^−4^ ± 4.0 × 10^−5^
V342M	dGTP	1.9 ± 0.1	126 ± 17	1.5 × 10^−2^ ± 2.1 × 10^−3^
K372N	dGTP	1.1× 10^−2^ ± 1.6 × 10^−4^	178 ± 8	6.2 × 10^−5^ ± 2.8 × 10^−6^

## 4. Discussion

Previous studies of disease-associated TERT variants have shown variable effects on activity and processivity. However, a comprehensive characterization of how disease-associated variants affect nucleotide insertion has been limited by enzymological and technical constraints. This study has determined the kinetic parameters of nucleotide insertion for six disease-associated TERT variants, as well as provided insight into the dynamic interactions of active site residues and incoming nucleotides (Table 2).

R194Q does not hydrogen bond with the triphosphate of the incoming nucleotide yet retains interactions with Q308. The decrease in its *k_pol_* and increase in *K_d_* suggests that the canonical role of R194 is to stabilize the incoming nucleotide rather than to participate in catalysis. A conserved arginine in DNA polymerase η has been implicated in the three-metal mechanism of nucleotide insertion [75]. Whether TERT utilizes a similar three-metal mechanism has not been determined, but future investigations following TERT throughout its catalytic cycle at higher resolution may reveal a similar mechanism used by DNA polymerase η. Losing both the stabilizing hydrogen bonds and the positive electrostatic charge in K372N explains the decreased nucleotide affinity demonstrated with our kinetic analysis. Our results support the role of K372 as a general acid during nucleotide insertion since K372N had a 95-fold decrease in its *k_pol_* [66]. K372N is still able to insert dGTP (albeit with dramatically decreased efficiency), which suggests alternative mechanisms exist for the protonation of the pyrophosphate in the absence of K372. Our MD simulations revealed that R340H altered sugar puckers for both the incoming nucleotide and the terminal 3′ nucleotide but had minimal changes in the overall position of the terminal 3′ nucleotide. Therefore, the altered sugar puckers of the incoming dGTP and the DNA 3′ terminus likely prevent efficient extension as was reported with other polymerases [62,72]. Interestingly, R865C and R865A hTERT mutations are also associated with TBDs, suggesting most mutations at this residue have pathological consequences [49,67].

Previous work has shown that V342M had minimal defects compared to WT, and the present work outlines the molecular basis of these changes [76]. The interactions we found with our MD simulations suggest V342M shifts the distribution of sugar puckers to the C3′-endo conformation which explains its increased rate of catalysis. The sidechain of V342M decreases the nucleotide binding pocket’s volume, causing the Watson–Crick hydrogen bonds between the incoming nucleotide and the templating base to lengthen compared to WT. The location of A255V is similar to that of polar filter residues found in some DNA polymerases [77]. Polar filters work in tandem with the steric gate residue by stabilizing the incoming nucleotide in an orientation that promotes steric clashing with the 2′-hydroxyl of an incoming rNTP and the steric gate [77]. However, A255V did not change the insertion efficiency of rGTP but rather decreased the efficiency of dGTP insertion by TERT. The most reasonable explanation for this finding is that A255 is located at a conserved position as reported polar filters but does not function as a polar filter. This is supported by the fact that while A255V disrupts hydrogen bonding with the incoming nucleotide, it did not alter rGTP insertion efficiency, as would be expected from disrupting the polar filter. Polymerase η also has an alanine at this position (A49 in hpolη) and it has been suggested polη has reduced sugar fidelity because it lacks a polar residue, such as threonine, at this position [77]. It is interesting to speculate that an A255T, or A255S TERT variant may have Increased sugar fidelity by establishing a polar filter in its active site. Y256N had similar binding affinities for dGTP and rGTP but a 10-fold decrease in its *k_pol_* for rGTP insertion. Our simulations suggest this mutant does not properly coordinate the triphosphate of rGTP. We observed the same phenomenon in our WT TERT simulations, but not in simulations of Y256A TERT, which has identical insertion efficiencies for both dGTP and rGTP [22]. The decreased sugar fidelity of A255V and Y256N may implicate changes in ribonucleotide insertion by telomerase have pathological consequences.

In summary, we have comprehensively characterized the nucleotide insertion mechanisms of six disease-associated TERT variants using pre-steady-state kinetics and MD simulations. Several of these variants are associated with cancer and understanding their nucleotide insertion mechanisms may allow for personalized telomerase targeted therapies. Various nucleotide analogs have been used to target telomerase and understanding how the WT enzyme utilizes these substrates will be crucial for designing more potent therapeutics. Furthermore, understanding how specific variants insert different nucleotide analogs can provide insight into specific molecules that may be more effective for treatment. For example, the decreased sugar fidelities of A255V and Y256N may increase the effective concentration of a nucleotide analog since they are able to insert both dNTP and rNTP versions of the molecule.

## Figures and Tables

**Figure 1 genes-14-00281-f001:**
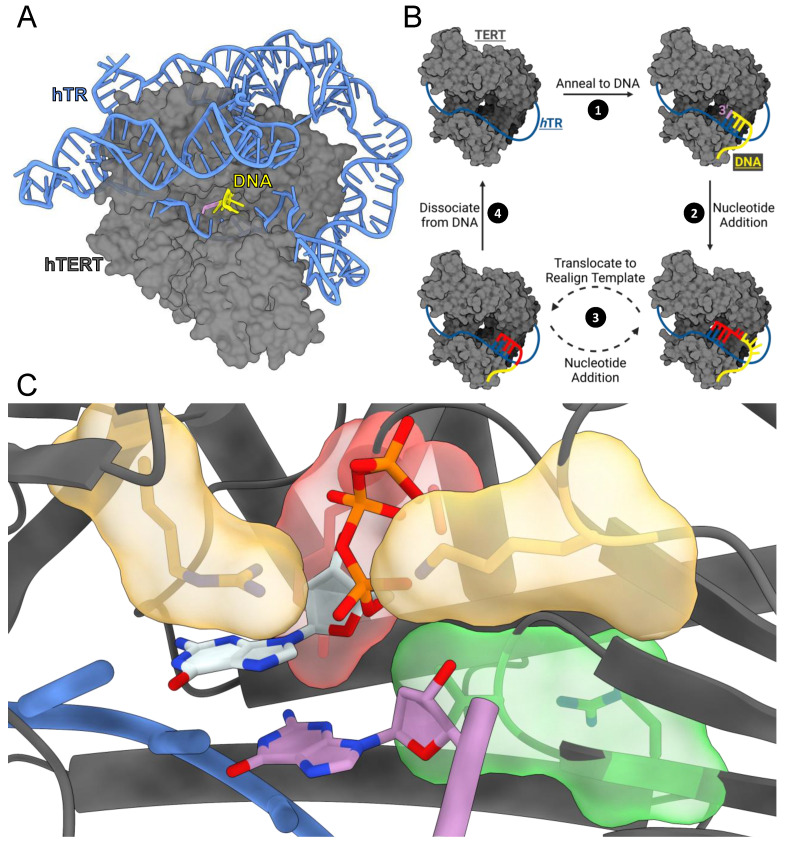
TERT structure and catalytic cycle. (**A**) hTERT/hTR (gray/blue) bound to DNA (yellow) with the DNA 3′ terminus indicating the active site (purple) (PDB: 7QXA). (**B**) Generalized diagram of telomerase’s catalytic cycle. (**C**) Closeup view of tcTERT active site with an incoming dGTP (PDB: 7KQN). TERT (gray), RNA template (blue), DNA 3′-terminus (purple), R194 and K372 (yellow), R340 and V342 (green), A255 and Y256 (red), and dGTP (white). Heteroatoms are depicted with CPK coloring (key: nitrogen (blue), oxygen (red), and phosphate (orange)).

**Figure 2 genes-14-00281-f002:**
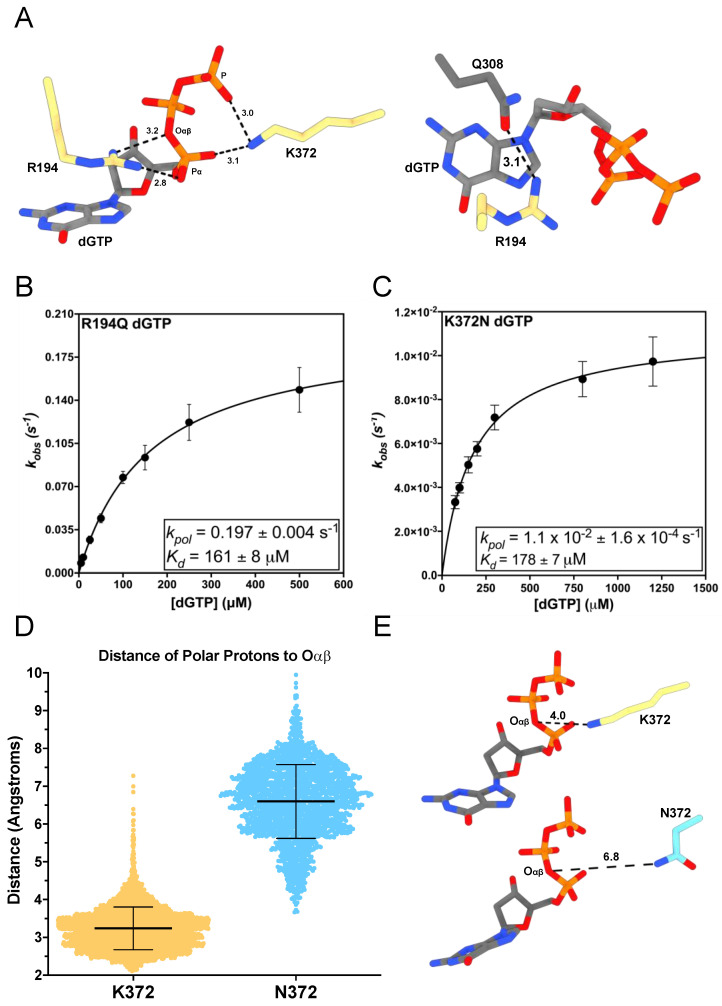
Mechanism of R194Q and K372N nucleotide insertion. (**A**) Interactions of R194 and K372 with the incoming dGTP and R194’s interaction with Q308 (PDB: 7KQN). (**B**) Replot of *k_obs_* values from single-turnover kinetics of R194Q inserting dGTP with the *k_pol_* and *K_d_* indicated. (**C**) Replot of *k_obs_* values for K372N inserting dGTP with the *k_pol_* and *K_d_* indicated. (**D**) Distances between the polar protons of K372 for WT (yellow) and N372 for K372N (cyan) and Oαβ of the incoming nucleotide. The average distance is indicated by the central black line and error bars represent the standard deviation of the mean. (**E**) Representative structures of WT and K372N, indicating the average positioning of the polar heavy atom for residue 372 in relation to Oαβ. The distance between the nonhydrogen atom closest to Oαβ for WT (yellow) and K372N (cyan) is represented with black dashes and measured in angstroms.

**Figure 3 genes-14-00281-f003:**
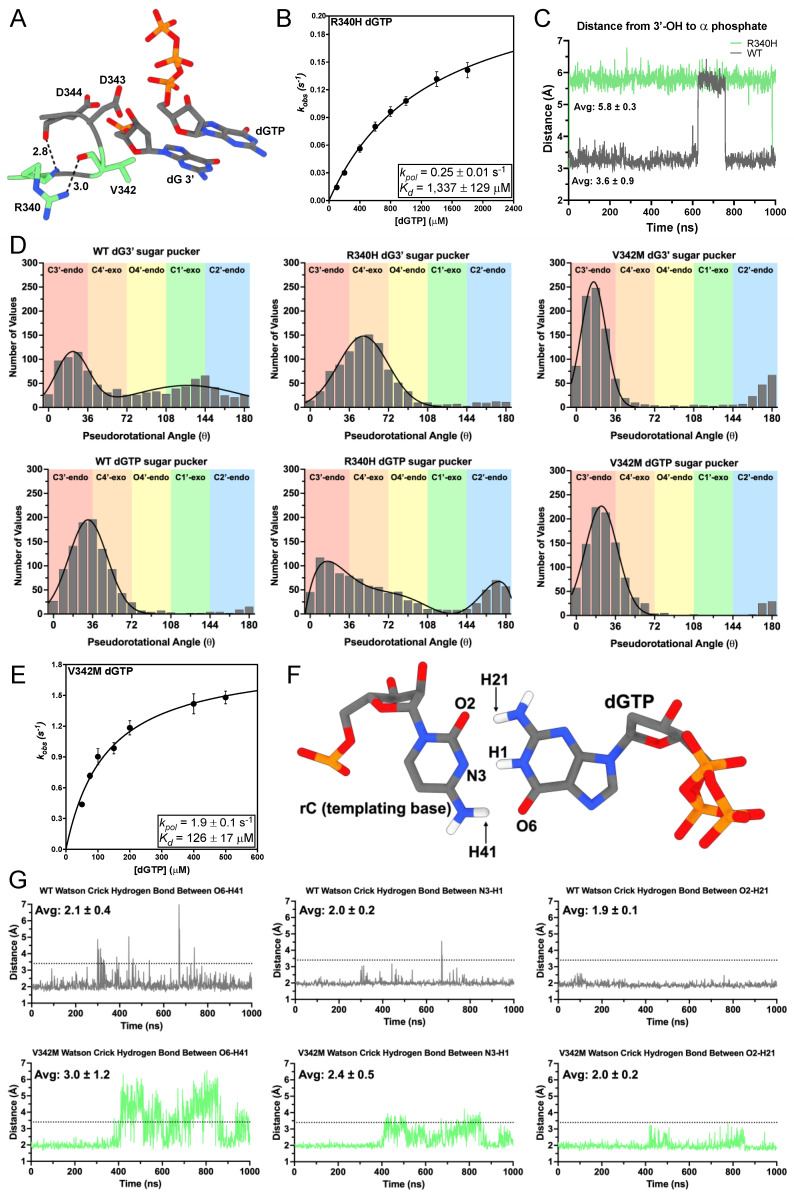
Mechanism of R340H and V342M nucleotide insertion. (**A**) Interactions between R340 and V342 with the 3′ terminal nucleotide (dG3′) and the incoming dGTP (PDB: 7KQN). (**B**) Replot of *k_obs_* values from single-turnover kinetics of R340H inserting dGTP with the *k_pol_* and *K_d_* indicated. (**C**) Distances between the 3′-hydroxyl of dG3′ and the ⍺-phosphate of dGTP for R340H (green) and WT (gray) ternary complexes. The average distance is indicated, and the error represents the standard deviation of the mean. (**D**) Pseudorotational angle distributions of the deoxyribose sugar for dG3′ and dGTP for WT, R340H, and V342M ternary complexes. Calculated values were binned (width 9°) and fit to either a single gaussian, sum of two gaussians, or polynomial function to identify local maxima. Sugar pucker conformations are defined at 36° intervals and are labeled according to their conformations and by color. (**E**) Replot of *k_obs_* values from single-turnover kinetics of V342M inserting dGTP with the *k_pol_* and *K_d_* indicated. (**F**) Atom identifiers involved in Watson–Crick hydrogen bonding between the templating base (rC) and the incoming nucleotide (dGTP). (**G**) Watson–Crick hydrogen bond distances for WT (gray) and V342M (green) over the course of the simulations. Average distances are indicated and error represents the standard deviation of the mean. A distance cutoff of 3.4 Å was used to determine hydrogen bonding and is indicated by the black dashed line.

**Figure 4 genes-14-00281-f004:**
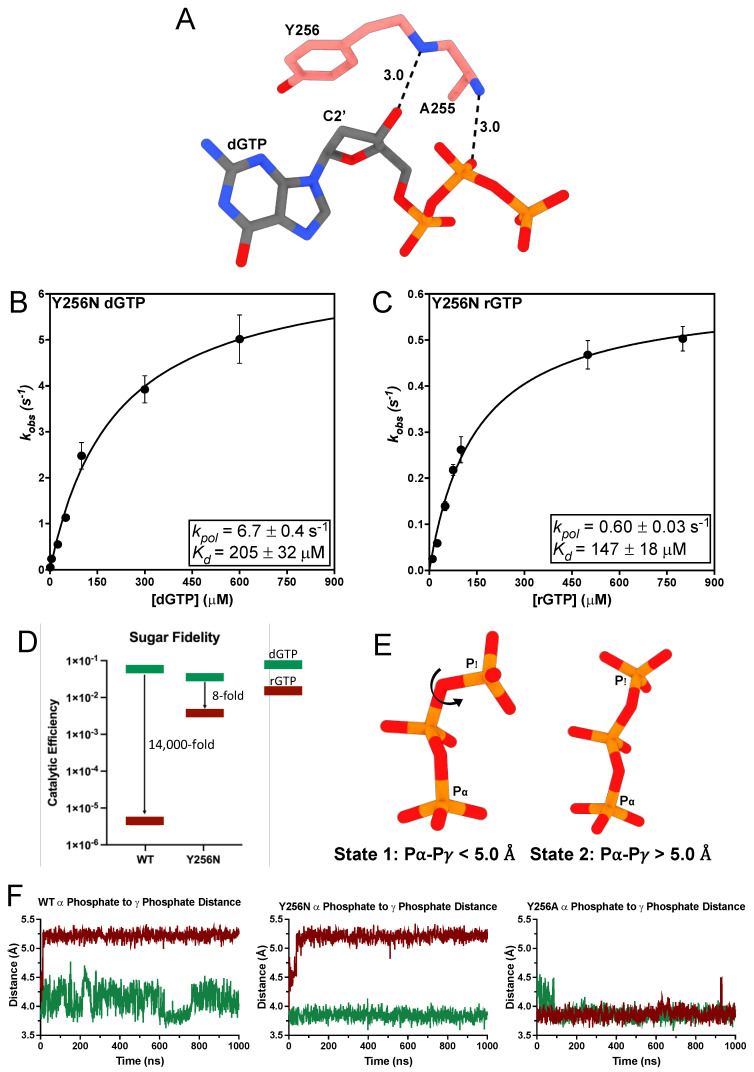
Mechanism of Y256N nucleotide insertion. (**A**) Interactions between A255 and Y256 with the incoming nucleotide (PDB: 7KQN). Hydrogen bond distances are represented with black dashes and measured in angstroms. (**B**) Replot of *k_obs_* values from single-turnover kinetics of Y256N inserting dGTP with the *k_pol_* and *K_d_* indicated. (**C**) Replot of *k_obs_* values from single-turnover kinetics of Y256N inserting rGTP with the *k_pol_* and *K_d_* indicated. (**D**) Sugar fidelity for WT and Y256N. WT values are taken from a previous publication [22]. (**E**) The two triphosphate states observed in the rGTP simulations. (**F**) Trajectories of the P⍺-P𝛾 distance for dGTP (green) and rGTP (red).

**Figure 5 genes-14-00281-f005:**
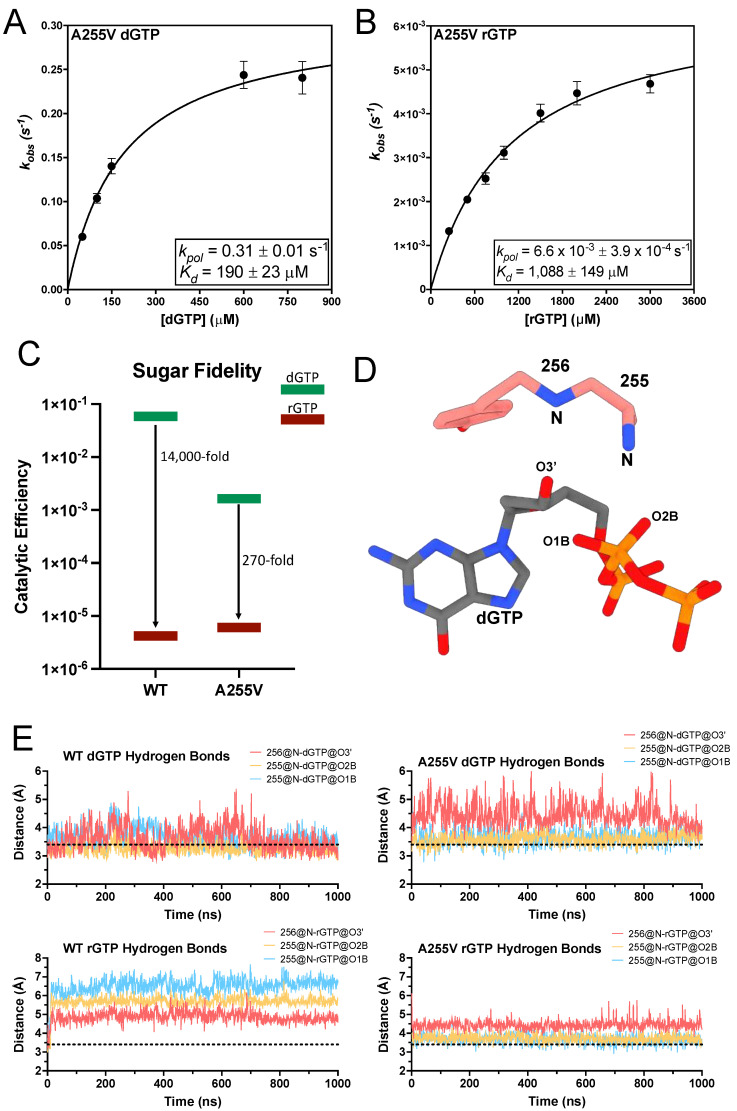
Mechanism of A255V nucleotide insertion. (**A**) Replot of *k_obs_* values from single-turnover kinetics of A255V inserting dGTP across from a templating rC with the *k_pol_* and *K_d_* indicated. (**B**) Replot of *k_obs_* values from single-turnover kinetics of A255V inserting rGTP across from a templating rC with the *kpol* and *K_d_* indicated. (**C**) Sugar fidelity for WT and A255V. WT values are taken from a previous publication [22]. (**D**) Atom identifiers involved in hydrogen bonds with the incoming nucleotide. (**E**) Trajectories of the distance between indicated atom pairs and dGTP or rGTP. A distance cutoff of 3.4 Å was used to determine hydrogen bonding and is indicated by the black dashed line.

**Table 1 genes-14-00281-t001:** hTERT disease-associated variants and homologous tcTERT variants investigated in this study.

hTERT Residue	tcTERT Residue	Disease Association	Reference
R631Q	R194Q	IPF	[43,44]
A716V	A255V	AA, IPF, Cancer	[45,46,47]
Y717N	Y256N	Cancer	[48]
R865H	R340H	AA, IPF	[49]
V867M	V342M	PF, DC	[43]
K902N	K372N	DC	[13]
D868	D343		
D869	D344		
Q833	Q308		

Note: (AA: Aplastic Anemia, IPF: Idiopathic Pulmonary Fibrosis, DC: Dyskeratosis Congenita). Orange residues were not directly investigated but formed important interactions with some variants. They are included here to assist the reader.

## Data Availability

The data presented in this study are available on request from the corresponding author.

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
