# Peer review of "Altered Nucleotide Insertion Mechanisms of Disease-Associated TERT Variants"

_genes, 2023, doi:10.3390/genes14020281_

Round 1

Reviewer 1 Report

The manuscript is technically sound and well-written. It highlights important results about the mechanism of nucleotide incorporation of six mutant forms of tcTERT that are associated with diseases in humans. However, major comments/suggestions are as follows:

·      The relationship between the six amino acid residues in tcTERT and hTERT are not well-defined. It would be helpful to address this by providing a sequence alignment of the TERT protein from at least these two organisms to show conservation of these “disease-associated” mutations. In addition, please label the location of each relevant amino acid within the structure of each protein (hTERT and tcTERT) in Supplemental Figure 1.

·      In addition to the locations of each of the six amino acids, Supplemental Figure 1 would also benefit by labeling each domain in the proteins (including the active site). If space permits, the location of the three residues shown in orange in Table 1 should also be indicated.

·      In Figure 1, the phosphate groups on the incoming dGTP are not indicated properly in the legend. Currently, it states that the DNA 3’ terminus is orange and the dGTP is white; however, the phosphate groups on the dGTP are orange too.

·      Although the six residues in the active site are indicated by color, it would be helpful if each of the specific amino acids was labeled well.

·      Page 6, line 237 error

·      Supplemental Figure 2 is not discussed in the Results section. Please mention this figure somewhere in the paper.

·      It would be very informative if STK was also performed using another nucleotide besides dGTP, since TERT can incorporate other nucleotides. At the very least, it should be mentioned as to why dGTP was chosen, and why the authors believe all studies using just one type of nucleotide incorporation is sufficient.

·      Much of the structural information explained in the Results section, especially referring to the MD simulations, was difficult to visualize as represented. The manuscript would benefit substantially from the inclusion of Figures of simulations from each of the six mutant binary/tertiary structures (at least a representative snapshot of each). All atoms mentioned in the simulation trajectories should be labeled within these MD simulation Figures too (for example, “R194@NH1-DGP617@O1B” in Supplemental Figure 3 and “256@N-DGP@O3’” in Figure 5D should be labeled to help visualize these specific atoms/bonds).

·      Figure 2E shows the average bond length between K372 and Oab is 4.0A, while the text (page 8, line 304) states this length is 3.2A. Same for N372 bond length.

·      Page 8, line 281 states “R194Q had a 5-fold decrease” but does not mention what parameter this 5-fold decrease refers to.

·      Please include a description for panel “F” in the legend of Figure 3.

·      The results/conclusions about different sugar puckering (Figure 3D and associated text) would be more convincing if the authors provided specific angle ranges for each endo/exo conformation mentioned in Figure 3D. It seems rather nebulous as currently written.

·      Please define the acronym “WC” on page 10, line 377.

·      Is there a typo in Figure 4E? Should the boxes be alphas? (P-box-P”)

·      The word “identical” is too strong of a word to use when comparing binding affinities between rGTP and dGTP by Y256N. They are similar, not identical.

·      The statement on page 12, lines 423-424 about the catalytic efficiencies of the Y256A mutant form of TERT for dNTP/rNTP should be referenced.

·      The last line in the second paragraph on page 12 (lines 425-426) is a little confusing. Is the coordination of the triphosphate “improper” for the Y256N mutant if it is similar to the coordination in the WT structure? And what is meant by “decreased catalytic rate?” Compared to Y256A? The rate of rGTP insertion by Y256N is higher than that by WT….

·      The paper would greatly benefit by the addition of a table listing all the STK data for WT and each mutant TERT protein. This would help organize the data and provide a reference for the Results and Discussion sections. In fact, the authors themselves confuse their own data at times (1. see point above, and 2. page 14 line 473 states: “decrease in both its kpol and Kd”, when the Kd is INCREASED).

·      The two functions of R194 are not clear on page 14, line 475. Please state the specific functions you are referring to.

Reviewer 2 Report

The authors of this study effectively deployed both experimental and computational modeling techniques to capture the biological impact of changes in key amino acid residues within the telomerase reverse transcriptase (TERT) protein. The authors thoroughly characterize several important amino acid residues in Tribolium Castaneum(tc) TERT that are involved in telomere lengthening catalyzed by the tcTERT complex. The authors adequately justify the use of tcTERT over the human TERT (hTERT) complex, given its structural similarities to hTERT and the availability of structural data to guide the downstream computational studies performed in this study.

Authors have focused on six disease-associated TERT variants. These six residues included ones that interact with the triphosphate of the incoming nucleotide, the sugar of the incoming nucleotide, and the 3’ terminus of the DNA. Using a series of single-turnover kinetics assays (STK), relative rate and dissociation constants are furnished for each of the six disease-associated TERT variants. The ratio of these two constants was then used to determine the relative catalytic efficiency of each mutation. The authors also leveraged the AMBER force field module to carry out molecular dynamic simulations on the same mutated residues to cross-validate the computationally simulated data with the experimentally determined values.

Results from the two methods - kinetic analysis and MD simulation - are complementary, and the strategy is ideally suited to providing insight into how mutations affect system function.  Overall, this study has succeeded admirably in investigating the biological impact and significance of disease-associated variants in TERT.

Specific Comments:

1, WT TERT data Kd and kpol values are cited in the text but not presented as data. Thus, it is unclear if the kinetic data of the TERT variants were compared with WT data produced from this study or the published values in their previous paper (Schaich et al, Elife 2022). Please clarify this.

2. Fig 2D:  What does the distribution of the distance means physically in the graph?

3. It would be helpful to discuss the validity of the MD method in predicting 3D structure.

4. It would be useful to present a table summarizing all the Kpol and Kd values of measured in this study,

5. The authors examined the kinetics of a single G insertion. Please discuss what would be possible outcome of the mutations in overall telomere lengthening, biochemically and at the cellular level.

6. Line 499: What does ‘polar filter residues’ mean?

7. In many places, fonts are incorrect or too small to recognize. (e.g. Fig 3D x-axis, Fig 4E, F)

8. Fig 3D:  Pseudo-rotational angle-sugar puckers labeling is unclear.

9. Fig 3F: Legend is missing.

10. It is unclear what several keys in the figures (e.g. PO-P” in Fig 4E;  256@N-DGF@ in Fig 5D, Sub Fig 3 ) are referring to.

11. There are spelling errors throughout the manuscript.

Reviewer 3 Report

Comments:

This is an interesting work where the authors have characterized kinetic factors involved in nucleotide insertion mechanisms for six disease-associated TERT variants employing steady state and MD simulation approaches. This work provides important information on how specific variant inserts various nucleotide analogs thus opening up avenues in designing potent therapeutics.

I have just have few minor comments.

Line 236-237: “R194 extends from above TERTs active site into hydrogen bonding distance of the 236 incoming nucleotides triphosphate (Error! Reference source not found. Is this a formatting error?

Figure 3: D and F the font size should be increased, as numbers are not clear on both the axis.
